# In Vitro Germination and Organogenesis of Endangered Neo-Endemic Baltic Dunes Species *Linaria loeselii* Schweigg

**DOI:** 10.3390/plants13172461

**Published:** 2024-09-03

**Authors:** Lidia Banaszczyk, Michał D. Starke, Damian Szelbracikowski, Julia Ścibior, Małgorzata Kapusta

**Affiliations:** 1Laboratory of Plant Cytology and Embryology, Department of Plant Experimental Biology and Biotechnology, Faculty of Biology, University of Gdańsk, 59 Wita Stwosza St., 80-308 Gdańsk, Poland; lidia.banaszczyk@ug.edu.pl (L.B.); michal.starke@ug.edu.pl (M.D.S.); 2Student Research Group of Biotechnology and Embryology “Explantatus”, Faculty of Biology, University of Gdańsk, 59 Wita Stwosza St., 80-308 Gdańsk, Poland; d.szelbracikowski.401@studms.ug.edu.pl (D.S.); j.scibior.046@studms.ug.edu.pl (J.Ś.); 3Bioimaging Laboratory, Faculty of Biology, University of Gdańsk, 59 Wita Stwosza St., 80-308 Gdańsk, Poland

**Keywords:** callus, environmental protection, micropropagation, ecosystems, biodiversity, organogenesis, plant tissue culture, Plantaginaceae, ex situ, in vitro

## Abstract

This study focuses on the endangered neo-endemic Baltic dunes species *Linaria loeselii* Schweigg. (Plantaginaceae), also known as *Linaria odora* (M. Bieb.). By utilizing in vitro cultures, we successfully germinated seeds collected in situ. Our method, which involved using media supplemented with 5 µmol/L 6-benzylaminopurine, led to the indirect regeneration of shoots after 60 days of culture in the dark, significantly increasing the number of progeny plants. Additionally, the medium supplemented with 2.85 μmol/L indole-3-acetic acid and 10.2 μmol/L paclobutrazol allowed rooting after 30 days of shoot fragments. This research provides a potential basis for developing *Linaria loeselii* introduction programs into the environment, thereby contributing to the conservation of this endangered species.

## 1. Introduction

Endemics are species that have adapted and naturally occur only in a specific geographic area, such as a continent, region, country, province, or even a small local area. The speciation of endemics is often associated with geographic barriers or unique ecosystems that require organisms to adapt strongly to the conditions there. Due to their strong adaptation to a particular habitat and limited occurrence, these species are particularly vulnerable to anthropogenic and natural environmental changes. This vulnerability underscores the importance of our research in the propagation of the endangered neo-endemic Baltic dunes species *Linaria loeselii* Schweigg., as a means of conservation [1].

Specific conditions around the Baltic Sea, such as dunes, have also led to the speciation of endemic species, including *L. loeselii* (Plantaginaceae), a neo-endemic species found exclusively between Unieście on the Lake Jamno sand bar in Poland, along the Vistula and Curonian Spit up to the Gulf of Riga in Latvia [2,3,4,5]. Phylogenetic analyses indicate that the species is closely related to the widely distributed *Linaria vulgaris* Mill.—a separate speciation of the two species occurred in the Quaternary about 250,000 years ago during glacial retreat [2,3,4,5,6]. Although closely related, these species differ significantly morphologically and in their habitats [7]. *L. loeselii* is sometimes described as *Linaria odora* (M. Bieb.) Fisch. or its subspecies (*Linaria odora* (M. Bieb.) Fisch. subsp. *loeselii*), but phylogenetic studies indicate that Baltic *L. loeselii* and Caspian *Linaria odora* are two different species [6,8,9].

*L. loeselii* occurs on white and gray dunes, occasionally on frontal dunes. It is also an element of migratory dunes, forming species-poor communities in deflation areas of eroded dunes and on sandy slopes. It is also often the only pioneer species found in open, dry areas of dunes [5]. *L. loeselii* is a plant with a rather loose habit, growing to about 50 cm tall with yellow fragrant flowers (Figure 1). This species is a pioneer geophyte with a root system of about 1 m in length and up to 30 m in extreme cases in the absence of competition, which favors the vegetative propagation of this species [10]. The roots of this species enter Arum-type arbuscular mycorrhiza, which is described as a characteristic of fast-growing plants [11]. *L. loeselii* is a photophilous species that prefers moderately warm climatic conditions and dry mineral-poor soil with neutral or alkaline reactions [10].

*L. loeselii* is an endangered species protected by Polish and international law, including the Berne Convention and the Conservation of European Wildlife and Natural Habitats Directive [12,13,14]. In Poland, it is also considered a species needing reintroduction efforts into the environment [15]. *L. loeselii*, like other endemic plants, is vulnerable to environmental transformation due to its considerable specialization in ecological conditions [1,15]. The main threat to *L. loeselii* is anthropopression. Since the sites of this species are also tourist attractions, plant trampling is a risk. The impact of tourism on dune areas can be mitigated by channeling tourist traffic or restricting access to precious areas by designating nature reserves or other forms of protection [5]. In the case of *L. loeselii*, as a pioneer species, dune stabilization through the creation of forest plantings to prevent dune erosion is also a threat. The disruption of the dynamic dune system leads to the disappearance of niches through occupation by other species. However, the conflict caused by dune stabilization and preservation from storm surges through coastal reforestation is largely unavoidable. Historical data show that wandering dunes can destroy cities closest to the coastal zone, demonstrating the need to compromise between protecting the population and conserving nature [16]. Also, threats from a warming climate, such as rising sea levels, the onset of violent and destructive weather events, or record heat, will pose an increasing threat to populations of *L. loeselii* in the future [17].

To preserve the species, it is essential to maintain a sufficiently large gene pool, giving the chance to keep as much genetic diversity as possible, which requires collecting material from specimen-poor sites as well [18,19]. Using in vitro cultures makes it possible to establish ex situ breeding from a minimal amount of starting material, which, in the case of species with a small number of individuals in situ, can ensure the possibility of securing an adequate gene pool while limiting environmental losses. In addition, the use of multiplication in the processes of organogenesis and somatic embryogenesis makes it possible to maximize the number of clonally obtained progeny plants. To date, in vitro culture methods have allowed the development of multiplication protocols and have realistically contributed to strengthening populations of some endemic and endangered plants [1,20,21,22]. The application of growth regulators, including auxins and cytokinins, has been demonstrated to significantly enhance callus formation and subsequent shoot regeneration in numerous studies involving species from the Plantaginaceae family. In particular, the combination of 6-benzylaminopurine (6-BAP) and 1-naphthalene acetic acid (NAA) has been demonstrated to promote callus induction, with optimal results observed at specific concentrations [23,24]. A study on the micropropagation of plant species in the Plantaginaceae family indicated that the supplementation of the 6-BAP substrate was associated with an increase in shoot growth. At the same time, adding the naphthalene acetic acid NAA affected the induction of roots on explants [23,24,25].

This study aimed to develop a method for rapid multiplication by in vitro propagation using indirect organogenesis. As a result, a method to preserve natural populations of *L. loeselii* will be developed. In addition, this protocol is the first to present the research on *L. loeselii* in this form.

## 2. Results

### 2.1. In Vitro Seed Germination

To determine the most favorable option for sterilizing *L. loeselii* seeds, seeds were surface sterilized with 70% ethanol after a 24 h incubation in a solution of gibberellic acid (GA_3_), followed by 5, 7, or 10 min in 1% sodium hypochlorite.

A 5 min sterilization with sodium hypochlorite proved insufficient to disinfect the seed husk thoroughly—most of the cultures became contaminated and were removed. Sterilizations of 7 and 10 min with sodium hypochlorite were sufficient to maintain the sterility of the cultures. It was noted that seeds sterilized for 10 min germinated faster than seeds sterilized for 7 min. As seedlings, we accepted seeds whose shells had been punctured by the root (Figure 2A). The seed first grew a shoot with cotyledons. After 38 days, all seeds from the 7 min and 10 min sterilizations had germinated.

After 4 days, most plants completed the seedling stage, producing the first leaves (Figure 2B). The seedlings obtained 40 days after germinating were transferred to substrates supplemented with activated carbon and continued under long-day conditions (16 h day/8 h night). After 90 days of the passage to the new substrates, the plants reached a size of about 10 cm, filling the culture container. The further culture was carried out by passages every 90 days to new substrates—2 cm shoots with preserved shoot apical meristem were used for passage—achieving high plant regeneration.

### 2.2. Induction of Indirect Organogenesis

*L. loeselii* leaf explants from plants from in vitro cultures were placed on media with varying concentrations of 1-naphthylacetic acid (NAA) and 6-benzylaminopurine (6-BAP) in the dark to multiply the amount of progeny material. After 30 days of culture, callus tissue was induced on explants, and the callus was cream-green, green, or cream-brown. Callus tissue differed slightly morphologically between the different variants of culture media (Figure 3).

The appearance of callus was observed on all variants of substrates. It was observed that adding NAA at both 1 and 5 µmol/L concentrations increased the frequency of explants on which callus appeared (Table 1). However, the best results were achieved with auxin media by adding cytokinin at a concentration of 1 or 5 µmol/L (Table 1). Similarly, the highest amount of callus was achieved with auxin and cytokine combination media, and there were no significant statistical differences between NAA and 6-BAP concentrations in these media. The appearing callus was green in color. On the control medium, the medium with the addition of 1 µmol/L 6-BAP with 5 µmol/L NAA, and on the medium with only 5 µmol/L 6-BAP, it was brighter and part white (Table 1). Callus induced on auxin-only medium, on medium with a high addition of cytokines, or with a high concentration of auxin and a low concentration of cytokines was compact. On the other media, callus with a brittle structure was observed (Table 1).

After another 30 days of callus culture, shoot organogenesis was observed (Figure 4). The highest frequency of explants on which organogenesis was induced was observed on a medium supplemented with 5 µmol/L 6-BAP and low-concentration or absent auxin (Table 2). Also, on the medium increased with 5 µmol/L NAA and 5 µmol/L 6-BAP, the frequency of explants on which induction occurred was high. However, the best medium for producing shoots was supplemented with 6-BAP alone at a concentration of 5 µmol/L.

### 2.3. Rooting

To increase the chance of plants from in vitro cultures acclimatizing, we evaluated the effect of IAA and paclobutrazol (PBZ) on the formation of seedlings that can be planted into soil culture. After four weeks, it was assessed that the medium that allowed the highest number of plants to root was the medium with 2.85 μmol/L IAA and 10.2 μmol/L PBZ. It was also evaluated that paclobutrazol significantly reduced the number of shoots produced and their length (Table 3, Figure 5).

## 3. Discussion

This study is a protocol that yields material in asymbiotic cultures that can protect selected individuals of *L. loeselii*, thus securing valuable gene pools. The effect of varying concentrations of NAA and 6-BAP on indirect organogenesis on *L. loeselii* leaves was also evaluated. We also tested the effect of IAA and PBZ on seedlings regenerated from shoot fragments for further work on acclimatization to soil culture.

Seeds collected in situ and sterilized with sodium hypochlorite were used as the starting material in the study. It was noted that seeds sterilized for 10 min germinated faster than seeds sterilized for 7 min. The longer sterilization time may indicate the effect of seed scarification on the germination rate—the positive impact of seed damage has been shown in many species, including the dune species *Penstemon haydenii* (Plantaginaceae)*,* in which mechanical damage to the shell by sand combined with cold stratification increased the number of germinated seeds [22]. A similar mechanism may occur under natural conditions in the case of *L. loeselii*.

The leaves of the in vitro-grown plants served as initial material for further studies on micropropagation using indirect organogenesis. Callus tissue formation was induced on all substrate variants after 30 days of experimentation, with the best results in terms of the amount of callus and its friable structure observed on substrates supplemented with 5 µmol/L 6-BAP with 1 or 5 µmol/L NAA. The most significant amounts of callus with compact structure were achieved on 1 µmol 6-BAP and 5 µmol/L NAA substrate. Also, the endangered representative of the Plantaginaceae family *Digitalis lamarckii* [23] or the medicinal species *Bacopa monnieri* [26] have been successfully propagated using tissue culture. These studies showed that a medium with auxins can be used for callus induction. In most cases, the MS medium supplemented with NAA alone or in combination with cytokinin can positively influence callus induction in this species [26]. Different concentrations of 6-BAP and NAA can lead to successful callus induction in other species belonging to the Plantaginaceae, such as *Plantago media* [27] or *Plantago major* [28]. Callus induction for these species can be observed after 14 days of culture [27], and represents a high percentage of callus formation (up to 100%) that occurred on the MS medium supplemented with different concentrations of the aforementioned auxins and cytokinins [27,28]. For *L. loeselii*, the induced callus was left on the induction medium for another 30 days to promote organogenesis. After 30 days of culture initiation, new shoots’ highest regeneration frequency was observed on a medium supplemented with 5 µmol/L 6-BAP. Also, Makowczyńska et al. showed that adding 6-BAP to the culture medium induces shoot organogenesis in *Plantago maritima* (Plantaginaceae) [24]. Similarly, Verma et al., in the case of *Digitalis lamarckii*, showed that NAA and 6-BAP substrate supplementation is sufficient to achieve effective shoot organogenesis [23]. In studies involving *Linaria genistifolia*, the application of 6-BAP has been demonstrated to effectively increase the number of shoots produced per explant, which is a crucial factor in the rapid propagation of the plant. The optimal concentration of 6-BAP has been shown to result in a significant increase in shoot multiplication, which facilitates the establishment of a larger number of plantlets from a limited amount of starting material [25]. In *L. loeselii*, it was noticeable that plants were directed toward rhizogenesis when supplemented with auxin. The structures obtained by indirect organogenesis were planted on a free-PGR medium and placed under long-day conditions, and plant regeneration was observed. In a study to develop a method for micropropagation of the endangered species *Artemisia arborescens* (Asteraceae), the source of explants was fragments of internodes. The highest average number of shoots was observed without auxin supplementation after adding 1.5 mg/L 6-BAP. In addition, in most variants of auxin and cytokinin concentrations, indirect adventitious organogenesis was induced, while direct organogenesis was kept on a medium supplemented with 0.5 mg/L 6-BAP [29]. In the case of research on *Swertia minor* (Gentianaceae), shoot induction depended on the type, concentration, and combination of plant hormones used. The MS medium not supplemented with plant hormones did not show shoot induction. In this study, the supplementation of the substrate with cytokinin 6-BAP alone contributed to a significant increase in shoot length in this species [30].

To prepare in vitro plants for soil culture conditions, shoot sections were inoculated onto IAA- and PBZ-supplemented medias for 30 days. Only the combination of 2.85 μmol/L IAA with 1.70 μmol/L PBZ yielded a significantly higher number of rooted shoots, considerably affecting the number of roots and their length. Exogenous auxin is commonly used for plant rooting and has also been used in studies for *Digitalis lamarckii* [23], *Plantago maritima* [24], and *Honckenya peploides* [31]. PBZ is a growth regulator that has multidirectional effects in plants, including altering the levels of gibberellins, abscisic acid, and cytokinins. The over-application of PBZ affects the activity of the photosynthetic system. It has also been noted to improve the resistance of plants to environmental stress so that it may find application in acclimatizing plants from in vitro to soil culture [32]. Our research has shown that applying PBZ significantly increased the number of roots in the seedlings. A similar effect was demonstrated by Kucharska et al. for chrysanthemums treated with PBZ, where an increase in root weight was observed [33]. It is possible that PBZ’s stimulation of endogenous abscisic acid production benefits root production in *L. loeselii*, as it does at some growth stages in *Arabidopsis thaliana*, *Medicago truncatula,* and *Oryza sativa* [34]. However, determining the exact mechanism requires further research.

In vitro culture methods have been used in many projects to receive valuable seedlings, including for *Salix lapponum* [35], *Aldrovanda vesiculosa* [36], and *Pulsatilla patens* [37]. These projects have preserved plant in vitro cultures and have culminated in the introduction of seedlings to rebuild populations, which is an undeniable benefit to conservation. The long-term effects of in vitro propagation on plant conservation strategies are complex and require careful consideration. Although in vitro techniques offer substantial benefits for the preservation of endangered species, considerable constraints and potential hazards can influence the efficacy of these strategies over time. In vitro propagation enables the expeditious multiplication of plant species, particularly rare or endangered ones. This can be of critical importance for the rapid increase in population size, particularly for species with limited natural reproduction rates [38]. The controlled environment of in vitro cultures minimizes the impacts of pests, diseases, and environmental stressors, thereby enhancing the survival rates of propagated plants. In vitro techniques, including cryopreservation, facilitate the long-term storage of genetic material from endangered species, thereby providing a genetic reservoir for future restoration efforts [39]. One of the primary concerns associated with in vitro propagation is the difficulty in translating laboratory success to natural environments. The absence of complex ecological interactions and abiotic factors present in the native habitats of plants grown in vitro may result in these plants’ lacking adaptive traits compared to their wild counterparts. This can result in difficulties when reintroducing these plants into the wild [40]. Consequently, future conservation strategies should integrate in vitro methods with comprehensive ecological assessments and community involvement. This will ensure that propagated plants can thrive in their natural habitats. This holistic approach is essential for the success of conservation efforts to preserve biodiversity in the face of ongoing environmental changes.

The propagation methods of *L. loeselii* presented in this paper can be used for ex situ conservation by securing individuals. The results provide a basis for further research on ex vitro acclimatization to prepare plants for reintroduction into the environment. Due to the peculiarity of the species, mainly occupying a dynamic coastal dune ecosystem, the process must be preceded by a thorough ecological analysis so that the result achieved is as beneficial to nature conservation as possible. Also, future studies of the seed banking of *L. loeselii* should be performed.

## 4. Materials and Methods

### 4.1. Plant Materials

The seeds, designated as the starting material, were procured from the yellow dune region on Hel (Pomeranian Voivodeship, E18.76909; N 54.65776) and subsequently integrated into the seed bank of the Department of Plant Taxonomy and Nature Conservation at the University of Gdansk.

### 4.2. Seeds Germination

To establish cultures of *L. loeselii*, the seeds were placed in syringes, which were closed with a nylon filter to prevent the seeds from falling out, according to the protocol outlined by [41]. The seed syringes were filled with 400 mg/L GA3 solution and placed in the refrigerator (4 °C) for 48 h to stimulate the seeds to germinate. After pretreatment, the GA3 solution was removed and replaced for 1 min with a 70% ethanol solution with a drop of Tween20. The seeds were placed for 5, 7, or 10 min in a 1% sodium hypochlorite solution (20% commercial ACE bleach solution), followed by rinsing with sterile distilled water three times. After sterilization, the syringe was opened by removing the nylon filter, and the seeds were poured into a Petri dish.

The seeds were transferred to the media with tweezers and needles. The germination medium contained half the salt and vitamins of Murashige and Skoog (M0222, Duchefa), supplemented with 2% *w*/*v* sucrose, and solidified with 7 g/L agar (A1262, Merck). Before adding the agar, the pH of the medium was set with NaOH and HCl at 5.8. The medium was autoclaved, after which it was bottled in plastic containers.

Cultures were conducted at 19 °C under long-day conditions (16 h day/8 h night) at a 100 μmol m^−2^ s^−1^ photosynthetic photon flux density (PPFD). The experiment was repeated twice. After leaving the seedling stage, germinated plants were transferred to a germination-like medium supplemented with 2 g/L activated carbon conducting cultures under the same conditions.

### 4.3. Shoot Organogenesis Induction

The media (with composition as for seed germination) prepared for the induction of shoot organogenesis were supplemented with various concentrations of NAA or 6-BAP after autoclaving. Leaf explants obtained from plants grown from seed in in vitro culture were used as the starting material for the study. The culture was carried out in the dark at 19 °C. The mean number of shoots and their length were documented at 30 and 60 days, limiting the effect of light on the explants. Five leaf explants were placed on each medium. The experiment was repeated five times.

### 4.4. Root Organogenesis Induction

To prepare plants that can be planted into soil culture, autoclaved media (with composition as for seed germination) were supplemented with varying concentrations of indoleacetic acid and paclobutrazol (PBZ). Shoot fragments approximately 2 cm in size were used as the starting material and placed on substrates. Cultures were conducted under long-day conditions (16 h day/8 h night) at a 100 μmol m^−2^ s^−1^ PPFD at 19 °C. Nine shoots were placed in each container, which constituted one repetition. The experiment was repeated three times. Cultures were conducted for 30 days, after which the mean number of roots and length were assessed.

### 4.5. Statistical Analysis

All data were analyzed using the GraphPad Prism statistical software package (version 10.0.2 for Windows) for ANOVA and Tukey tests (*p* ≤ 0.05).

## 5. Conclusions

The presented method of in vitro culture and multiplication may find application in the ex situ conservation of *L. loeselii*. In addition, the developed method may strengthen or restore populations of this endangered and biodiversity-valuable species by using the resulting cuttings or seeds for wild introduction programs. Using the techniques of in vitro culture allowed us to establish an efficient protocol for the ex situ propagation of the valuable species *L. loeselii* from seeds comprising the following steps: sterilization of seeds in 1% NaOCl for 10 min; germination on ½ MS medium; multiplication of seedlings on basal MS medium complemented with plant growth regulators (5 µmol/L 6-BAP); and rooting the multiplicated plants on MS complemented with 2.85 µmol/L IAA and 10.2 µmol/L PBZ.

## Figures and Tables

**Figure 1 plants-13-02461-f001:**
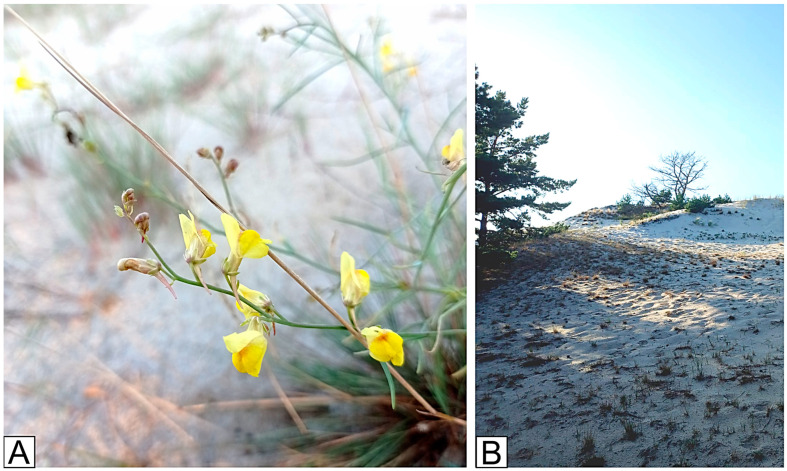
*Linaria loeselii* Schweigg. (**A**) Plant in situ. (**B**) Hel dunes reserve—one of the natural sites of *L. loeselii* on the Polish coast of the Baltic Sea.

**Figure 2 plants-13-02461-f002:**
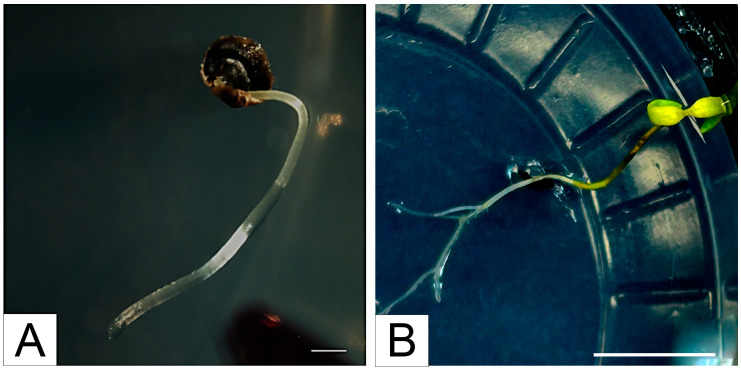
Seed germination and in vitro seedling development of *Linaria loeselii* Schweigg. (**A**) A germinating seedling pierces the seed husk of a seed collected from in situ plants (bar indicates 1 mm). (**B**) Young plants from seed collected from in situ plants with first leaves developed (bar indicates 1 cm).

**Figure 3 plants-13-02461-f003:**
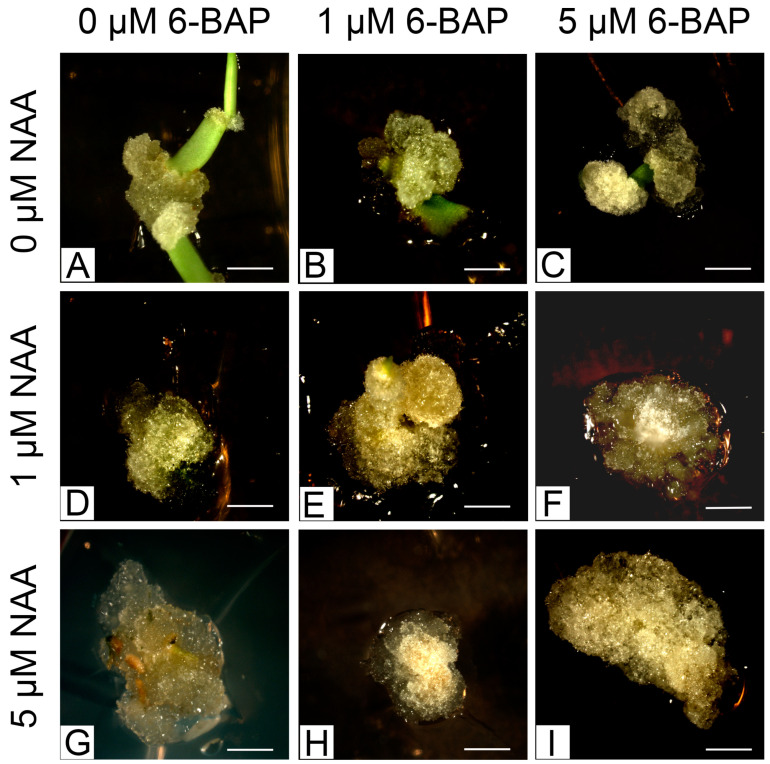
Comparison of callus tissue morphology on media supplemented with different concentrations of 6-BAP and NAA: (**A**) control medium (no growth regulators); (**B**) 1 μmol/L 6-BAP; (**C**) 5 μmol/L 6-BAP; (**D**) 1 μmol/L NAA; (**E**) 1 μmol/L 6-BAP and 1 μmol/L NAA; (**F**) 5 μmol/L 6-BAP and 1 μmol/L NAA; (**G**) 5 μmol/L NAA; (**H**) 1 μmol/L 6-BAP and 5 μmol/L NAA; (**I**) 5 μmol/L 6-BAP and 5 μmol/L NAA (all bars indicate 1 mm).

**Figure 4 plants-13-02461-f004:**
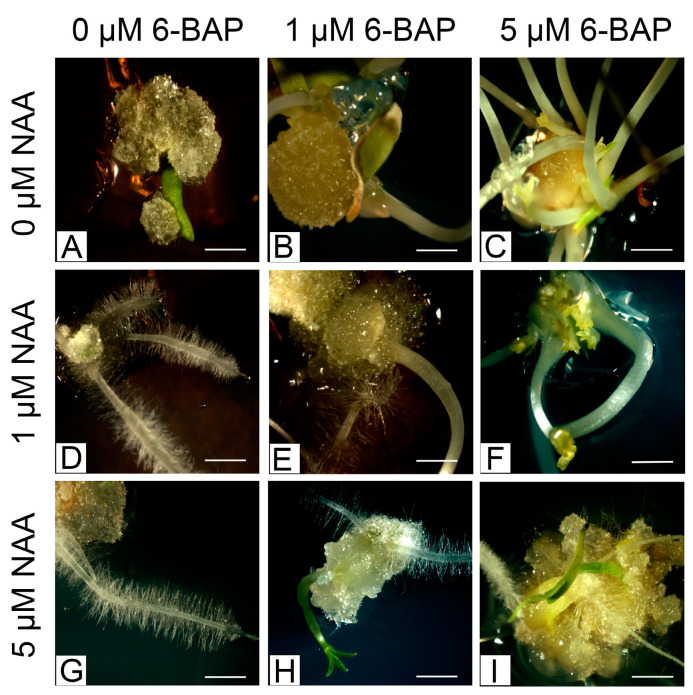
Induction of indirect organogenesis on media supplemented with different concentrations of 6-BAP and NAA: (**A**) control medium (no growth regulators); (**B**) 1 μmol/L 6-BAP; (**C**) 5 μmol/L 6-BAP; (**D**) 1 μmol/L NAA; (**E**) 1 μmol/L 6-BAP and 1 μmol/L NAA; (**F**) 5 μmol/L 6-BAP and 1 μmol/L NAA; (**G**) 5 μmol/L NAA; (**H**) 1 μmol/L 6-BAP and 5 μmol/L NAA; (**I**) 5 μmol/L 6-BAP and 5 μmol/L NAA (all bars indicate 1 mm).

**Figure 5 plants-13-02461-f005:**
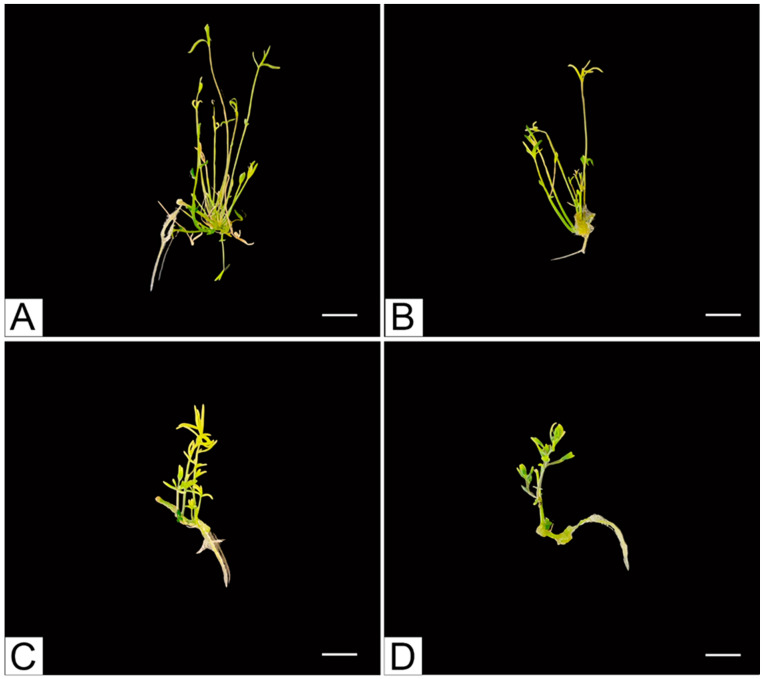
Shoot regeneration and rooting of *Linaria loeselii* Schweigg. shoots on IAA- and PBZ-supplemented media after 30 days of culture. (**A**) Control medium (no growth regulators); (**B**) 2.85 μmol/L; (**C**) 2.85 μmol/L IAA and 1.70 μmol/L PBZ; (**D**) 2.85 μmol/L IAA and 10.2 μmol/L PBZ (all bars indicate 1 cm).

**Table 1 plants-13-02461-t001:** Effect of ½ MS medium supplemented with different concentrations of 6-BAP and NAA on callus induction after 30 days of culture. Data are expressed as the mean of five independent bioassays with *p* < 0.05 (Tukey’s post hoc test). The same letters indicate no significant statistical differences between variants.

PGRs Concentration		After 30 Days			
6-BAP (μmol/L)	NAA (μmol/L)	Mean CallusInduction (%)	Mean Callus Size (In mm^2^)	Structure	Color
0	0	34 a	4.07 ± 0.6 a	friable	white-green
0	1	82 b	28.79 ± 2.5 b	compact	green
0	5	74 b	26.43 ± 2.5 b	compact	green
1	0	44 a	5.66 ± 0.9 a	friable	green
1	1	90 c	43.21 ± 2.7	friable	green
1	5	98 c	53.71 ± 2.5 c	compact	cream-green
5	0	46 a	9.95 ± 1.4 a	compact	white-green
5	1	97 c	55.99 ± 2.5 c	friable	green
5	5	94 c	53.51 ± 2.7 c	friable	green

**Table 2 plants-13-02461-t002:** Effect of 1/2 MS medium supplemented with different concentrations of 6-BAP and NAA on induction of indirect organogenesis after 30 days of culture. Data are expressed as the mean of five independent bioassays with *p* < 0.05 (Tukey’s post hoc test). The same letters indicate no significant statistical differences between variants.

6-BAP (μmol/L)	NAA (μmol/L)	Mean Induction of Shooting (%)	The Mean Number of Shoots	Mean Shoot Length (mm)
0	0	15 a	1.88 ± 0.31 b	3.4 a
0	1	12 a	2.00 ± 0.37 b	3.9 a
0	5	6 b	1.67 ± 0.31 b	3.7 a
1	0	12 a	2.36 ± 0.37 b	5.4 ab
1	1	22 c	1.75 ± 0.26 b	5.2 ab
1	5	14 a	1.75 ± 0.21 b	3.1 a
5	0	31 c	5.14 ± 0.37 a	10.94 b
5	1	27 c	1.50 ± 0.16 b	6.3 ab
5	5	16 a	2.92 ± 0.60 b	3.5 a

**Table 3 plants-13-02461-t003:** Effect of 1/2 MS medium supplemented with different concentrations of IAA and PBZ on rooting and shooting after four weeks of culture. Data are expressed as the mean of three independent bioassays with *p* < 0.05 (Tukey’s post hoc test). The same letters indicate no significant statistical differences between variants.

IAA (μmol/L)	PBZ (μmol/L)	Mean Induction of Rooting (%)	The Mean Number of Roots	Mean Roots Length (mm)	The Mean Number of Shoots	Mean Shoot Length (mm)
0	0	11 a	1.00 ± 0.0 ab	1.7a	3.33 ± 0.6 a	29.63 a
2.85	0	22 a	1.42 ± 0.20 ab	5ab	2.81 ± 0.4 ab	25 ab
2.85	1.70	56 b	1.33 ± 0.16 b	8.9 b	1.56 ± 0.2 bc	19.2 ab
2.85	10.2	40 ab	1.9 ± 0.10 a	6.3 ab	1.15 ± 0.1 c	16.9 b

## Data Availability

Data are contained within the article.

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
