# Peer review of "In Vitro Germination and Organogenesis of Endangered Neo-Endemic Baltic Dunes Species Linaria loeselii Schweigg"

_plants, 2024, doi:10.3390/plants13172461_

Round 1

Reviewer 1 Report

Comments and Suggestions for Authors

This is an interesting and in some ways pioneer study on micropropagation of the endemic and endangered species Linaria loeselii. The species occupies a very specific ecological niche formed by dunes in several European countries, particularly in the Baltic, where it is included in national and regional Red Books and was designated as endangered.

The subject fits into the scope of the journal and the special issue. The experiments are relevant for the purpose of the study and, in general, are well done, followed by the straightforward and clearly interpreted results. In my opinion, the paper will be quite useful for researchers and conservation specialists working on the conservation of the European flora.

That been said, I believe that the tissue culture propagation protocol suggested in the study is incomplete without the experiments on ex vitro acclimatization of the in vitro-rooted plants. Acclimatization is usually a critical step for such protocols and is absolutely necessary for the reintroduction of the species back to natural environment. I would highly recommend extending the research by performing simple acclimatization experiments and presenting their results, even if not 100% successful.  

I also list below a couple of suggestions for improving the presentation of the manuscript:

Introduction: Seed banking is the easiest and usually the most effective, in terms of the gene diversity coverage, approach to conserve endemic species. Why it was not tested for Linaria loeselii? Please mention if the species could be propagated by seeds and why the authors did not consider this option (seeds are dormant? not viable? difficult to collect?).

Also in Introduction: Please justify using these specific growth regulators (BAP, NAA) for propagation. Maybe they were effective for micropropagation of the relative species?

Figure 2. Legend. Consider changing to “Seed germination and in vitro seedling development”.

Figures 3, 4. Legends. (A-I) No need to repeat the same text every time, it is intuitively understandable. Consider changing to the following: “(A) control (no growth regulators); (B) 1 µmol/l BAP; (C) 5 µmol/l BAP…” etc.

A reference to Table 2 is missing in the text (or I couldn’t find it).

Tables 1, 2 and 3. The first row with “PGRs concentration” and “After 30 days” (or “after 4 weeks”) can be removed. Move “after 30 days” or “after 4 weeks” to the table title instead.

All tables. The decimal point is not a comma but a period, please check in all table and correct when needed.

Table 2. The maximum mean number of shoots reported here is 1.241 which is actually one shoot per explant. Most treatments showed 0.2-0.4 shoots per explant on average which means no shoots at all. Based on these numbers, the protocol is underdeveloped, not effective and requires further optimization. I believe that these numbers are the results of counting shoots formed on all explants. Maybe it would be more informative if you count a number of shoots per explant only for leaf fragments with morphogenesis.   Same for root number in Table 3 – count root number only on explants with rhizogenesis.

Table 1 (Mean number of shoots) and Table 3 (Mean number of roots and shoots) - two decimal places are enough.

Lines 161-162. “..the best medium for producing many shoots…”. In fact, your best result is 1.241 shoots which is, in average, one shoot, not “many”. See my comment above for potential solution.

Line 268. “The method… make it possible to obtain plants that can be planted..” – Acclimatization experiments are required to prove this statement.

Line 272. “The presented method of in vitro breeding…” – there were no breeding experiments, please remove.

Materials and methods: - please add information about the parameters evaluated. For example, for 5.3, this could be “mean callus induction, mean number of shoots per explants and mean shoot length were counted after 30 and 60 days of the experiment….”

Discussion. Please include studies on the tissue culture propagation of relative species belonging to the same genera, if any.

Please consider English editing service or have an English-native colleague look at the manuscript. Some sentences look like an automatic translation, and there are gramma errors in the text.

Comments on the Quality of English Language

The manuscript is, generally, well-written, but I would still recommend a style editing or a help from an English-native colleague. Some sentences, although understandable, look like an automatic translation. There are grammatical errors as well.

Author Response

Dear Reviewer,

thank you for taking the time to review our manuscript. We appreciate your thoughtful feedback and effort. Your insights have significantly contributed to improving the quality of our work.

We have carefully considered your suggestions and made the following corrections based on your input:

Comments 1: Introduction: Seed banking is the easiest and usually the most effective, in terms of the gene diversity coverage, approach to conserve endemic species. Why it was not tested for Linaria loeselii? Please mention if the species could be propagated by seeds and why the authors did not consider this option (seeds are dormant? not viable? difficult to collect?).

Response 1: The research conducted for Linaria loeselii is part of a larger project; it involved the establishment of tissue culture, which we present in the article, and the creation of a seed bank. Our research focused on obtaining plants from seeds collected from the environment for a maximum of one year before sowing. Unfortunately, we do not have data on the decline in viability of Linaria loeselii seeds during storage for extended periods. Data are available on another species from the Plantaginaceaea family, Synthyris bullii. They showed extremely low seed germination after about ten years of seed storage. Researching the viability of Linaria loeselii seeds after long-term storage is undoubtedly a valuable topic. However, it's important to note that this was a different focus from the subject of the presented research. The experiment aimed to obtain as many plants as possible from a single seed rather than studying the viability of seeds after long-term storage, which is crucial in seed banking. However, we thank the reviewer for his valuable comment regarding the Seed Bank. We will use this suggestion in planning our next years' research.

Nevertheless, storage in in vitro cultures of Linaria loeselii is possible through regular passages. In addition, just as we indicate, it is possible to support multiplication by supplementing the medium with auxins and cytokinins.

Comment 2: Also in Introduction: Please justify using these specific growth regulators (BAP, NAA) for propagation. Maybe they were effective for micropropagation of the relative species?

Response 2: Thank you for this suggestion. In the introduction, we outlined the impact of plant growth regulators (NAA and 6-BAP) on in vitro cultures in plants belonging to the Plantaginaceae family. This change can be found in page 3 (lines 83-91).

Comment 3: Figure 2. Legend. Consider changing to “Seed germination and in vitro seedling development”.

Response 3: We agree with this comment. It was done (page 4, line 102).

Comment 4: Figures 3, 4. Legends. (A-I) No need to repeat the same text every time, it is intuitively understandable. Consider changing to the following: “(A) control (no growth regulators); (B) 1 µmol/l BAP; (C) 5 µmol/l BAP…” etc.

Response 4: Thank you for this suggestion. We have corrected figures descriptions as suggested.

Comment 5: A reference to Table 2 is missing in the text (or I couldn’t find it).

Response 5: Corrected. “Table 3” is erroneously used twice in the text. We have corrected this (page 7, line 155).

Comment 6: Tables 1, 2 and 3. The first row with “PGRs concentration” and “After 30 days” (or “after 4 weeks”) can be removed. Move “after 30 days” or “after 4 weeks” to the table title instead.

Response 6: Thank you for this suggestion. We have corrected tables headings and titles.

Comment 7: All tables. The decimal point is not a comma but a period, please check in all table and correct when needed.

Response 7:  Thank you for notice this. We have corrected that in all tables.

Comment 8: Table 2. The maximum mean number of shoots reported here is 1.241 which is actually one shoot per explant. Most treatments showed 0.2-0.4 shoots per explant on average which means no shoots at all. Based on these numbers, the protocol is underdeveloped, not effective and requires further optimization. I believe that these numbers are the results of counting shoots formed on all explants. Maybe it would be more informative if you count a number of shoots per explant only for leaf fragments with morphogenesis.   Same for root number in Table 3 – count root number only on explants with rhizogenesis.

Response 8: Thank you for this solution. Exactly, these numbers are the results of counting shoots formed on all explants.  We corrected into shoots per explants.

Comment 9: Table 1 (Mean number of shoots) and Table 3 (Mean number of roots and shoots) - two decimal places are enough.

Response 9: Thank you for this suggestion. We corrected that.

Comment 10: Lines 161-162. “..the best medium for producing many shoots…”. In fact, your best result is 1.241 shoots which is, in average, one shoot, not “many”. See my comment above for potential solution.

Response 10: Corrected, like in response 8.

Comment 11: Line 268. “The method… make it possible to obtain plants that can be planted..” – Acclimatization experiments are required to prove this statement.

Response 11: Dear reviewer, we are grateful for these suggestions. We also think the critical point of growing plants in vitro culture is their ex vitro acclimatization.

When it comes to acclimatisation, it is not so easy for two reasons. The first point is we need a suitable place to do the acclimatisation, while the place is currently occupied by other crops from other experiments, other researchers. The second, second point is that this is a plant a protected plant, so we need permission from the relevant authority to extend the study and perform the acclimatisation.

During our research, we initiated the first attempt to acclimatize plants from in vitro conditions. The preliminary results are promising, but we are still testing additional factors that will help us optimize the acclimatization method for this species, given its demanding adaptations to environmental conditions. Consequently, we intend to improve this protocol further and prepare a separate manuscript.

Comment 12: Line 272. “The presented method of in vitro breeding…” – there were no breeding experiments, please remove.

Response 12: We removed this sentence and we improved the beginning of conclusions. We hope that in this version of manuscript now it’s enough.

Comment 13: Materials and methods: - please add information about the parameters evaluated. For example, for 5.3, this could be “mean callus induction, mean number of shoots per explants and mean shoot length were counted after 30 and 60 days of the experiment….”

Response 13: Thank you for this suggestion. We have corrected that (page 12, lines 340 and 352).

Comment 14: Discussion. Please include studies on the tissue culture propagation of relative species belonging to the same genera, if any.

Response 14: We added information about in vitro propagation of close relatives species. In introduction and discussion part.

Comment 15: Please consider English editing service or have an English-native colleague look at the manuscript. Some sentences look like an automatic translation, and there are gramma errors in the text.

Response 15: Thank you for your suggestion. We have done our best to correct grammatical errors, but once the changes have been accepted by all reviewers, the final version of the paper will undergo language correction once again.

Reviewer 2 Report

Comments and Suggestions for Authors

The authors have studied the in vitro propagation of the endangered neo-endemic Baltic dunes species Linaria loeselii Schweigg, focusing on three main aspects: in vitro seed germination, induction of indirect organogenesis, and rooting. For the induction of indirect organogenesis, the authors compared callus tissue morphology on media supplemented with different concentrations and assessed the induction of indirect organogenesis on media with various concentrations. Callus tissue was successfully induced from explants after 30 days, with optimal results achieved using a combination of auxins and cytokinins in the culture media. The highest frequency of shoot organogenesis was observed on media supplemented with 5 µmol/L 6-BAP, indicating effective conditions for plant regeneration. For rooting, the optimal medium contained 2.85 µmol/L IAA and 1.70 µmol/L PBZ, which significantly enhanced root production. Additionally, the study highlighted the effectiveness of paclobutrazol (PBZ) in influencing root development, suggesting its potential application in acclimating in vitro plants to soil conditions.

Comments:

1. However, the study lacks complete images of shoot regeneration and rooting. Please provide complete images of both shoot regeneration and rooting processes.

2. Include a comprehensive analysis of the genetic diversity present in the regenerated plants, which could limit the understanding of how well the in vitro propagation methods can capture the genetic variability necessary for long-term conservation efforts.

3. The observations made in the study are primarily short-term, focusing on initial growth and regeneration rates.

4. The long-term effects of in vitro propagation are not discussed, which is crucial for assessing the success of conservation strategies.

Comments on the Quality of English Language

Minor editing of English language required.

Author Response

Dear Reviewer,

thank you very much for your review of our manuscript. Your constructive feedback has been incredibly helpful, and we appreciate your time and effort.

In response to your comment, we have made following adjustments:

Comment 1: However, the study lacks complete images of shoot regeneration and rooting. Please provide complete images of both shoot regeneration and rooting processes.

Response 1: We added images showing shoot regeneration and rooting, which are included in Figure 5.

Comment 2: Include a comprehensive analysis of the genetic diversity present in the regenerated plants, which could limit the understanding of how well the in vitro propagation methods can capture the genetic variability necessary for long-term conservation efforts.

Response 2: Thank you very much for this valuable comment. We realize that plants obtained in vitro culture may undergo processes leading to somaclonal variation. Expanding our research to include tests of genetic variation in plants obtained through micropropagation would undoubtedly increase the utility of our proposed methods. Unfortunately, we do not have the resources to conduct this type of research.

Realizing that obtaining plants devoid of artificially induced genetic changes is essential for conservation purposes, breeding was carried out in a way that limited the possibility of such changes occurring. We used regular passages and did not use agents such as amino acids that excessively stimulate plant growth.

Comment 3: The observations made in the study are primarily short-term, focusing on initial growth and regeneration rates.

Response 3: We agree. We clarify the title of the manuscript by replacing the word “propagation” with “germination and organogenesis.”.

Comment 4: The long-term effects of in vitro propagation are not discussed, which is crucial for assessing the success of conservation strategies.

Response 4: We added this information into discussion part (page 11, lines 273-292).

Comment 5: Minor editing of English language required.

Response 5: Thank you for your suggestion. We have done our best to correct grammatical errors, but once the changes have been accepted by all reviewers, the final version of the paper will undergo language correction once again.

Reviewer 3 Report

Comments and Suggestions for Authors

Comments and Suggestions for Authors

This study focuses on the in vitro propagation of Linaria loeselii Schweigg. (Plantaginaceae), an endangered neo-endemic of the Baltic dunes. Linaria loeselii, like other endemic plants, is vulnerable to environmental transformation due to its considerable specialization in ecological conditions. In this regard, the present study of the possibilities of in vitro reproduction of the species is relevant and provides a potential basis for its introduction into the environment, thus contributing to the conservation of this endangered species.

The manuscript is arranged according to requirements of “Plants”. The applied research design and research methods are appropriate to achieve the purpose of the study. The results are well presented and supported with good quality tables and figures. Furthermore, the results were processed through quantitative and statistical analysis.

The following remarks and suggestions can be made:

-          The full name of the studied species - Linaria loeselii Schweigg. (Plantaginaceae), is used only in its first citation, and in continuation only the short name - L. loeselii - should be used.

Introduction

-          In this part many information on the species habitats is given that is redundant. It is enough to mention that it is endemic, in which normative documents it appears as protected and briefly the state of its populations. Тhere no sufficient information on the application of in vitro methos in related species (of the same genus). It should also be emphasized that this is the first study of its kind with the target species.

-          The aim need to be redacted, as follows: This study aimed to develop a method for rapid multiplication by in vitro propagation using technic of indirect organogenesis. In result a method to preserve natural populations of Linaria loeselii will be developed.

Materials and Methods

-          I suggest the following redaction of the point ‘5.1. Plant materials ‘: The starting material consisted of L. loeselii seeds from the yellow collected from dune area on Hel (Pomeranian Voivodeship, E18.76909; N 54.65776). The seeds came from the seed bank of the Department of Plant Taxonomy and Nature Conservation at the University of Gdansk.

Discussion

-          The part between lines 254-264 is more appropriate for the Introduction

Conclusions

-          The last part of this section /lines 275-281/ must be redacted, as follows: Using the technics of in vitro culture allowed establish an efficiency protocol for ex situ propagation of the valuable species Linaria loeselii from seeds comprising the following steps: sterilization of seeds in 1% NaOCl for 10 min; germination on ½ MS medium; multiplication of  seedlengs on basal MS medium complemented with plant growt regulators (5 µmol/l 6-BAP); rooting the multiplicated plants on MS complemented with 2,85 µmol/l IAA and 1,7 µmol/l PBZ.

 In conclusion, this manuscript is recommended for publication in “Plants”.

Author Response

Dear Reviewer, 

we appreciate your thorough review of our manuscript. Your feedback has provide us with a clear path to improve our work. 

We made the following revisions: 

Comment 1: The full name of the studied species - Linaria loeselii Schweigg. (Plantaginaceae), is used only in its first citation, and in continuation only the short name - L. loeselii - should be used.

Response 1:  Thank you for this suggestion. We corrected that.

Comment 2: In this part many information on the species habitats is given that is redundant. It is enough to mention that it is endemic, in which normative documents it appears as protected and briefly the state of its populations. Ð¢here no sufficient information on the application of in vitro methos in related species (of the same genus). It should also be emphasized that this is the first study of its kind with the target species.

Response 2:  We added information about in vitro propagation of close relatives species (page 10, lines 233-238). But we find only one research including this kind of research in Linaria spp.

Comment 3: The aim need to be redacted, as follows: This study aimed to develop a method for rapid multiplication by in vitro propagation using technic of indirect organogenesis. In result a method to preserve natural populations of Linaria loeselii will be developed.

Response 3: Done (page 3, lines 92-95).

Comment 4: I suggest the following redaction of the point ‘5.1. Plant materials ‘: The starting material consisted of L. loeselii seeds from the yellow collected from dune area on Hel (Pomeranian Voivodeship, E18.76909; N 54.65776). The seeds came from.

Response 4: Thank you for your suggestion. We corrected that (page 12, lines 312-315).

Comment 5: The part between lines 254-264 is more appropriate for the Introduction.

ODP: We place those sentences into introduction part (page 2, lines 58-72).

Comment 6: The last part of this section /lines 275-281/ must be redacted, as follows: Using the technics of in vitro culture allowed establish an efficiency protocol for ex situ propagation of the valuable species Linaria loeselii from seeds comprising the following steps: sterilization of seeds in 1% NaOCl for 10 min; germination on ½ MS medium; multiplication of  seedlengs on basal MS medium complemented with plant growt regulators (5 µmol/l 6-BAP); rooting the multiplicated plants on MS complemented with 2,85 µmol/l IAA and 1,7 µmol/l PBZ.

Response 6: Thank you for this suggestion. We change that sentences (page 12, lines 304-309).

Round 2

Reviewer 2 Report

Comments and Suggestions for Authors

The authors have addressed my comments.